# Genomic and virulence analysis of *in vitro* cultured *Cryptosporidium parvum*

**Nigel Yarlett**[1,2]*, **Mary Morada**[2], **Deborah A. Schaefer**[3], **Kevin Ackman**[3], **Elizabeth Carranza**[3], **Rodrigo de Paula Baptista**[4,5¤], **Michael W. Riggs**[3], **Jessica C. Kissinger**[4,5,6]

1 Department of Chemistry and Physical Sciences, Pace University, New York, New York, United States of America, 2 Haskins Laboratories, Pace University, New York, New York, United States of America, 3 School of Animal and Comparative Biomedical Sciences, University of Arizona, Tucson, Arizona, United States of America, 4 Center for Tropical and Emerging Global Diseases, University of Georgia, Athens, Georgia, United States of America, 5 Institute of Bioinformatics, University of Georgia, Athens, Georgia, United States of America, 6 Department of Genetics, University of Georgia, Athens, Georgia, United States of America

☯ These authors contributed equally to this work.
¤ Current address: Houston Methodist Research Institute, Houston, Texas, USA
* nyarlett@pace.edu

## Abstract

Recent advances in the *in vitro* cultivation of *Cryptosporidium parvum* using hollow fiber bioreactor technology (HFB) have permitted continuous growth of parasites that complete all life cycle stages. The method provides access to all stages of the parasite and provides a method for non-animal production of oocysts for use in clinical trials. Here we examined the effect of long-term (>20 months) *in vitro* culture on virulence-factors, genome conservation, and *in vivo* pathogenicity of the host by *in vitro* cultured parasites. We find low-level sequence variation that is consistent with that observed in calf-passaged parasites. Further using a calf model infection, oocysts obtained from the HFB caused diarrhea of the same volume, duration and oocyst shedding intensity as *in vivo* passaged parasites.

## Author summary

*Cryptosporidium parvum* and *C. hominis* are waterborne parasites that are the second to third leading cause of diarrheal disease, and a major contributor to childhood deaths worldwide. Traditionally these intestinal parasites have proven difficult to culture for more than 2–3 days, which hampers long term in vitro studies. We reasoned that cultures of intestinal epithelial cells as monolayers in static plates results in the production of unpolarized epithelial cells. Utilizing hollow fiber technology, we have developed a method for producing intestinal epithelial cell growth that simulates the body resulting in polarized intestinal epithelial cells that have basal and apical surfaces, tight junctions, and develop functional villi. Using this system, we have maintained in vitro cultures of *C. parvum* that produce all life cycle stages for 20 months. Long-term in vitro culture of parasites often results in the development of a phenotype that is no longer pathogenic to the host; In this publication we show that using a calf model *C. parvum* BGF-T20HF after 20 months of in vitro culture was unchanged with respect to diarrhea output, parasite load,

**Editor:** Ana Maria Cevallos, Universidad Nacional Autonóma de México Instituto de Investigaciones Biomédicas: Universidad Nacional Autonoma de Mexico Instituto de Investigaciones Biomedicas, MEXICO

**Data Availability Statement:** All data needed to evaluate the conclusions of this paper are presented in the paper, its supplementary materials or deposited in an online database. Nucleotide sequences have been deposited in the GenBank under BioProject PRJNA877237 (SRA accessions: SRR24474866, SRR24474867). BGF-2017 raw Illumina reads are available under SRA accession SRR11516703.

**Funding:** Financial support was provided by The Bill and Melinda Gates Foundation awards OPP1117675 (N.Y.), OPP1151701 (J.C.K), Investment 44418 from the Bill and Melinda Gates Foundation (M.W.R.). Investment GH VAP NG-ID20 from Bill and Melinda Gates Foundation (NY). The funders had no role in study design, data collection and analysis, decision to publish, or preparation of the manuscript. M.M., R.P.B., and D. S., received salary support from the BMGF.

**Competing interests:** The authors have declared that no competing interests exist.

and clinical scores from the isolate used to initiate the culture (BGF-T0). In addition, we can show that the genome of the cultured parasites (BGF-T20HF) has undergone a similar genomic drift as the parent isolate (BGF-T0) used to start the inoculum that had been maintained by passage through fetal calves. Collectively the data supports the use of the in vitro cultured isolate, BGF-T20HF, for human trials, and provides a long-term model for the development of novel chemotherapeutic drugs to treat this disease.

## Introduction

Diarrheal disease is a major cause of worldwide morbidity and mortality, particularly amongst children and immunocompromised individuals. Childhood diarrhea alone is estimated to be responsible for 800,000 deaths annually [1]. Data obtained by the global enteric multicenter study [2] demonstrate that most cases can be attributed to rotavirus, *Cryptosporidium sp*, Enterotoxigenic *Escherichia coli*, or *Shigella*. *Cryptosporidium parvum* and *C. hominis* are the causative agents of human cryptosporidiosis, a moderate-to-severe diarrheal disease (MSD), that was reported to be the second to third leading cause of diarrheal disease in under 23-month-old pediatric cases in low socio-economic areas, where it is estimated to result in 7.6 million cases and 202,000 deaths annually [3]. In contrast to higher socio-economic areas, those with low-economic resources had a high incidence of pediatric deaths due to contaminated drinking water supplies resulting from unsanitary conditions and lack of suitable water purification systems [1]. Whereas in higher socio-economic areas, outbreaks of cryptosporidiosis are commonly associated with recreational water supplies such as swimming pools, water parks, hot tubs, and spas. Death results from severe dehydration, and survivors often have long term effects resulting in malnutrition, stunted growth, and cognitive impairment [4]. Current recommended therapy for the disease is nitazoxanide, which is not approved for use in children under 12 months of age, which is the population most vulnerable to the effects of cryptosporidiosis [5]. Advances in the molecular biology and *in vitro* culture of the parasite [6–11] have provided an impetus to the drug discovery platform resulting in several promising leads [12–15]. The inability to obtain a complete sexual life cycle of the parasite during *in vitro* growth has hindered biochemical and molecular studies. Conventional 2D cultures do not provide the structural or environmental requirements to permit continuous growth of the parasite and results in asexual reproduction that fails to perpetuate the infection beyond 2–3 days [11]. The use of the hollow fiber bioreactor (HFB) facilitates the creation of a three-dimensional culture system, where the basal surface of the intestinal epithelial cells are attached to the outer surface of the porous hollow fibers thus permitting access to nutrients and oxygen flowing inside the fibers or the intracapillary space (ICS), and the apical surface of the epithelial cells differentiate to form a multilayered polarized surface on the outside of the fibers or the extra-capillary space (ECS). Using this method, we have developed a nutrient rich, low redox ECS medium that replicates the lumen of the gut allowing *C. parvum* to be maintained in continuous *in vitro* culture for >20 months, during which time oocysts were transferred to secondary and tertiary HFB where they continued to grow. We have previously shown that oocysts from the HFB were infective *in vivo* using several mouse models [16]. However, the *C. parvum* mouse model provides limited information on the pathology of the infection, as the mouse model fails to develop diarrhea, hence the severity of the mouse model infection is based upon reduced weight gain compared to control mice, and numbers of oocysts shed. The ability of the HFB to generate large numbers ($5 \times 10^7$/mL) of axenic oocysts that do not require chlorination or treatment with strong oxidizing agents is an advantage for biochemical and

medical studies using oocysts. However, *in vitro* cultivation of cells involves an adaptation process that can result in a population of parasites exhibiting significant differences to the original wild-type inoculum [17]. These adaptive responses can negatively impact virulence factors and result in significantly reduced infections in animal models particularly after long term *in vitro* culture [18–20]. Additionally, many studies have observed that *in vitro* cultured parasites lose virulence factors and have limited use as models of infection [17]. Hence it was the goal of this study to evaluate *C. parvum* oocysts generated by the HFB *in vitro* culture method for potential changes in their repertoire of genes, the presence of insertions and deletions (InDels), and changes in Single Nucleotide Variants (SNV's). We also compared the clinical outcome of HFB *in vitro* cultured oocysts with those obtained from calf-passaged oocysts in the calf model [21], which together with the gnotobiotic piglet model [22] are the primary models demonstrating clinical symptoms including significant and sustained diarrhea due to cryptosporidiosis.

## Materials and methods

### Ethics statement

All *in vivo* studies were carried out in strict accordance with the recommendations in the *Guide for the Care and Use of Laboratory Animals* of the National Institutes of Health [23]. The calf model protocol (09–120) was approved by the Institutional Animal Care and Use Committee of the University of Arizona, Tucson, AZ (Animal Welfare Assurance number A-3248-01). Calf studies were performed in compliance with guidelines in the Animal Welfare Act and *Guide for the Care and Use of Agricultural Animals in Research and Teaching* [24]. The animal biosafety level 2 (ABSL-2) facilities used were fully accredited by the American Association for Laboratory Animal Care. All efforts were made to minimize suffering of animals employed in these studies.

### Origin of *C. parvum* IOWA isolates used in this study

BGF-T0 was purchased from Bunch Grass Farms in March 2016 and sequenced at the Beijing Genomics Institute (BGI). BGF- 2017 was purchased from Bunch Grass Farms and sequenced by the GGBC at the University of Georgia. BGF-T20HF–Originated from BGF-T0 following 20 months of continuous culture in the HFB and was sequenced by BGI. IOWA-ATCC–Genomic DNA was ordered from ATCC (catalog number ATCCPRA-67DQ) and sequenced at the Wellcome Sanger Institute (WSI) as in [25]. IOWA-2017 was produced at the *Cryptosporidium* Production Lab (University of Arizona).

### Hollow fiber culture of *C. parvum*

*C. parvum* was cultured using a hollow fiber bioreactor [8] containing a 20 kD MW cut off polysulfone fiber cartridge (FiberCell Systems, Frederick, MD). HCT-8 cells were grown on the extra capillary surface of the fibers, and nutrients provided to the basal cell surface from the intracapillary space which contained minimum essential media plus 10% horse serum (MEM + 10% HS) and the following supplements: 0.058 g/L heparin, 0.29g/L L-glutamine, 23.8 g/L HEPES pH 7.8, 4.5 g/L D-glucose, 0.035 g/L ascorbic acid, 0.04 g/L p-aminobenzoic acid, 0.02 Ca pantothenate, 0.001 folic acid. The intracapillary space medium was pumped at 2.5 mL/min from a 1L reservoir. The extra capillary space contained the following supplements dissolved in MEM + 10% HS: 135 mg/L taurodeoxycholate, 4.5 mg/L thioglycolic acid, 49.5 mg/L mannitol, 3.0 mg/L each of glutathione, taurine, betaine, and cysteine, plus 1.34 mg/mL oleic acid, and 3.6 mg/L cholesterol. When the glucose concentration of the intracapillary

space dropped to 50% (2.25 g/L) in 24 h, $10^6$ *C. parvum* oocysts were inoculated into the extra capillary space as previously described [16].

## *C. parvum* Enumeration

Samples (2 mL) were removed from the HFB and total RNA was isolated from pellets obtained by centrifugation at 6449 x g for 5 min (Beckman-Coulter, Indianapolis, IN, USA) using iScript RT-qPCR sample preparation kit (Bio-Rad Labs, Hercules, CA, USA) as previously described [8]. Total RNA was obtained using RNeasy (Qiagen Inc, Valencia, CA, USA) and quantitated using a Qubit 3.0 fluorometer (Life Technologies, Thermo-Fisher Scientific Inc., Waltham, MA, USA). *C. parvum* 18S rRNA was amplified by qRT-PCR using an iScript One-Step qRT-PCR kit with SYBR green (Bio-Rad Labs) containing specific primers for *C. parvum* 18S-rRNA (*Cp*18S-995F: 5′-TAGAGATTGGAGGTTCCT-3′ and *Cp*18S-1206R: 5′-CTCCAC-CAACTAAGAACGCC-3′). Total RNA, reagents and primers were incubated at 48˚C for 30 min, followed by 95˚C for 10 min, and subjected to 40 cycles of 95˚C for 15 s and 60˚C for 1 min. A melting curve was performed by heating to 95˚C for 15 s, followed by 60˚C for 15 s and 95˚C for 15 s, using a Quant Studio 6 flex Real-Time PCR system (Life Technologies, Thermo-Fisher Scientific Inc, Waltham, MA, USA). Oocyst standards of $10^5$, $10^6$, and $10^7$ oocysts were included, and parasite numbers were evaluated from a graph of the log *C. parvum* oocysts versus $C_T$ for the parasite SSU rRNA [25]. This method was selected since RNA has a turnover time of hours compared to DNA and hence is a better indication that we are enumerating live parasites. For genomic and calf-model infections oocysts were purified using Dynabeads anti-*Cryptosporidium* (Thermo Fisher Scientific, Waltham, MA) as described using the manufacturers' instructions.

## Fluorescent labelled antibody staining

Oocysts and motile stages from the HFB were observed using antibody-specific fluorescent dye conjugates. Briefly, 1 mL sample from the HFB was centrifuged at 16,162 x g for 1 min (Sorvall Biofuge Fresco, Thermo Fisher Scientific) and the pellet resuspended in 100 μL PBS. Oocysts were stained using 25 μL each of an FITC labelled mouse monoclonal antibody to *C. parvum* oocyst surface proteins (Crypt-a-Glo, Waterborne Inc., New Orleans LA, USA) and a fluoresceien-labelled polyclonal antibody (Sporo-Glo, Waterborne Inc., New Orleans, LA, USA) to motile stages, for 30 min in the dark, washed twice with PBS and examined using a fluorescence microscope (Nikon Eclipse Ts2R inverted fluorescence microscope) with an excitation wavelength of 410–485 nm and an emission wavelength 515 nm (Crypt-a-Glo), or an excitation wavelength of 535–550 nm, and an emission wavelength 580 nm (Sporo-Glo).

## Sequencing protocol

BGF-T0 was acquired from Bunch Grass Farms in March 2016 and underwent sequencing at the Beijing Genomics Institute (BGI) using their BGISEQ DNBSEQ-G400 short-read paired-end sequencing platform. BGF-2017, obtained from Bunch Grass Farms, was subjected to pair-end short-read sequencing by the GGBC at the University of Georgia, employing the Illumina MiSeq sequencer. BGF-T20HF, originating from BGF-T0 through 20 months of continuous cultivation in the HFB, was also sequenced at BGI using the same short-read platform as BGF-T0. For IOWA-ATCC, genomic DNA was procured from ATCC (catalog number ATCCPRA-67DQ) and subsequently sequenced at the Wellcome Sanger Institute (WSI) as described in reference [22]. The production of IOWA-2017 took place at the *Cryptosporidium* Production Lab (University of Arizona). The mean coverage levels for BGF-T0, BGF-T20F, and BGF-2017 were determined to be 1421x, 60x, and 237x, respectively.

## Variant call analysis

Illumina short-read sequences from (BGF-T0, BGF-T20HF and BGF-2017) were aligned to *C. parvum* IOWA-ATCC v50 [25] available at CryptoDB.org [26] with the addition of three extra sub-telomeric regions for chromosomes 7 and 8 (GenBank Accessions:MZ892386-8) using BWA-mem v0.7.17 [27]. The alignments were then submitted to Picard Toolkit v2.16.0 [28] to parse these alignments and remove duplicates. The genome analysis Tool Kit v4 (GATK4) [29] Haplotype Caller was used to call variants. All variants were filtered using GATK Variant Filtration with the following parameters: phred quality > 30, depth > 10, mapping quality > 40 and fisher statistics < 60.0. The final filtered variant call file (vcf) was then annotated using SNPeff v5.1 [30] using the *C. parvum* IOWA-ATCC v50 genome annotation as a custom database.

## Comparative sequence analysis

All annotated vcf files were submitted to SnpSift extractFields v5.1 [31] to generate variant tables. They were then compared using BedTools v2.29.2 [32] intersect and subtract to identify the shared and unique variants for each sample. Plots were generated using Venny v2.1 [33].

## Specific Gene alignments

To generate gene sequences for each isolate, we used a *de novo* assembly approach using SPAdes v3.15 [34], followed by a reference guided scaffolding approach using Ragtag v2.1 [35], and annotation transfer using Liftoff v1.6.1 [36]. Protein sequences were generated using AGAT v0.4.0 [37] and aligned using MAFFT v7 [38]. To determine if the called variants were affecting protein domains, all target genes were submitted to InterProScan v5 [39].

## Calf model infection

A total of 9 calves (3 per group) were obtained from the same United States Department of Agriculture (USDA)-licensed closed-herd dairy vendor [21]. Calves were fed commercial colostrum replacer within 2 h after birth (bovine IgG colostrum replacement; Land O'Lakes, Shoreview, MN) per label instructions. Triplicates were randomly assigned using the Microsoft Excel random number generation tool (Redmond, WA) to be infected with BGF-T20HF (oocysts collected from the HFB culture), BGF-2017 (a recent sample obtained from Bunch Grass Farms), and IOWA-2017 (the parent isolate for BGF routinely calf passaged at the University of Arizona (Fig 1B). Experimental personnel were blind to isolate assignments during the study. All calves were housed in an ABSL-2 facility in separate containment rooms. Precautions and disinfection measures were taken for the deliveries and housing of these calves to prevent unintended *Cryptosporidium* or other enteropathogen infection. The calves were fed antibiotic-free milk replacer (Nutrena Snowflakes calf milk II-Utiliz milk replacer; Cargill Animal Nutrition, Minneapolis, MN) twice daily from 12 h of age until termination of the experiment at day 10 postinfection (PI). An oral electrolyte solution (Re-Sorb; Pfizer) was supplemented once diarrhea developed in an animal. At 36 to 48 h of age (study day 0), each calf was infected by oral inoculation of $5 \times 10^7$ purified disinfected *C. parvum* oocysts (BGF-T20HF, BGF-2017, and IOWA-2017). Stool samples were collected every 24 h starting on study day 3 PI. The total volume of feces excreted for successive 24-h collections was recorded. Total daily oocyst counts for each calf were determined as previously described [40]. Briefly, qPCR was used to quantify *C. parvum* oocysts from feces collected over successive 24-h periods which had been well mixed using a commercial blender to ensure sample uniformity. Calves were also assigned numerical scores for the following variables twice daily: clinical

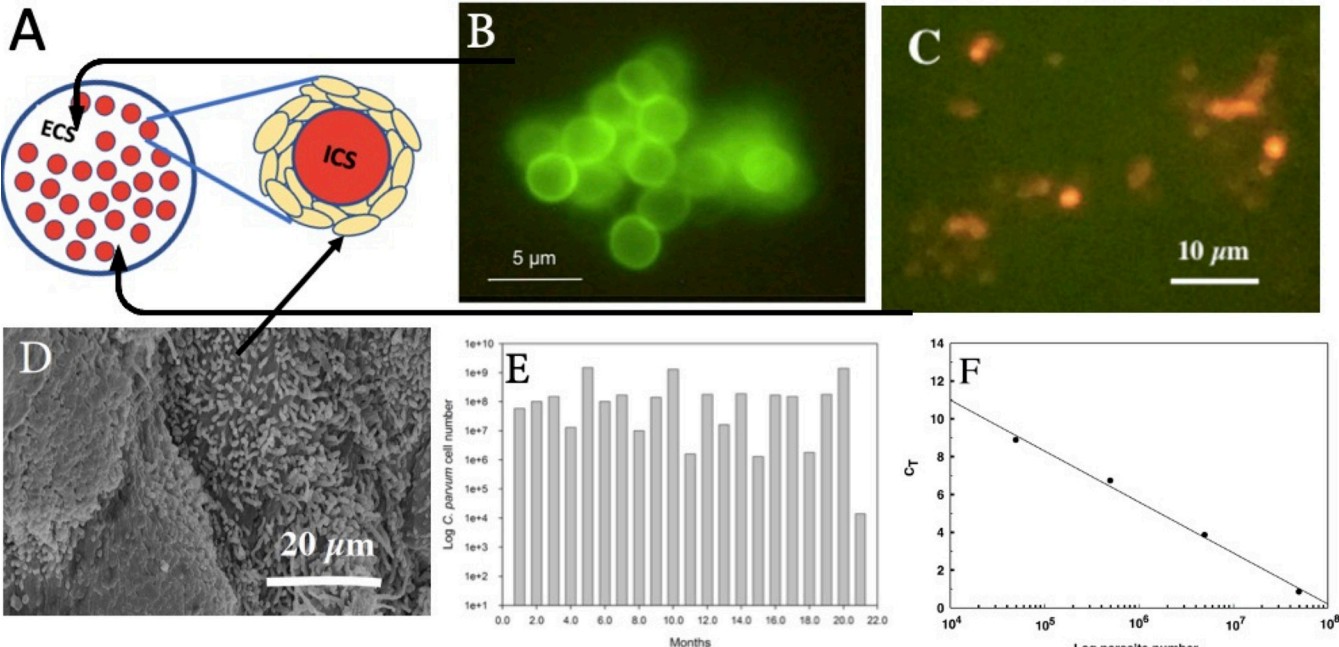

**Fig 1. Diagramatic section through the hollow fiber bioreactor.** The cartridge contains a series of hollow fibers through which the host cell growth media, MEM plus supplements and 10% horse serum is pumped through the intracapillary space (ICS). The extracapillary space (ECS) contains the host epithelial cells (HCT-8) that attach to and grow around the outside of the fibers forming a 3D matrix that receives nutrients from the basal surface. *C. parvum* is inoculated into the ECS after the host cell 3D layer has developed as determined by a drop in the ICS glucose concentration of 50% or more in 24h. *C. parvum* sporozoites attach to the apical surface of the host epithelial cells as they do in the intestinal tract. (A) Diagrammatic section through the HFB showing the extracapillary space (ECS) and host cells growing around the intracapillary space (ICS). (B) *C. parvum* oocysts from the HFB stained with Crypt-a-Glo. (C) Sporo-Glo stained merozoites and sporozoites from the HFB. (D) Electron microscope image of the HCT-8 cells grown on the fibers showing the presence of microvilli; obtained by sectioning a 3-month cartridge. (E) Growth of *C. parvum* based upon qRT-PCR of samples collected from the HFB during the 20-month culture period as described in the methods. (F) Typical $C_T$ plot used to quantitate *C. parvum* growth.

symptoms, general health (willingness to rise, stance, rectal temperature, appetite and food intake, attitude, and hydration status), presence or absence of diarrhea, and fecal consistency [41]. All calves were euthanized on study day 10 PI.

## Results

### *Cryptosporidium parvum* culture

*In vitro* cultures of *C. parvum* (IOWA isolate) were generated using an inoculum of $10^6$ oocysts supplied by Bunch Grass Farms (BGF-T0) and maintained for 20 months using the HFB culture method [8,16] (Fig 1). The culture produced approximately $10^7$–$10^8$ oocysts/mL when sampled every 7–10 days (HFB produced oocysts are referred to as BGF-T20HF; Fig 2).

### Calf infections

Infectivity and clinical scores from the calf clinical model for cryptosporidiosis were obtained for the *C. parvum* isolate after 20 months of continuous culture in the HFB (BGF-T20HF) and compared with calf-passaged isolates from the same supply BGF-2017 in addition to *C. parvum* IOWA-2017, which was maintained at the University of Arizona (Table A in S1 Table). Infectivity and clinical scores were obtained from days 3–10 PI (Fig 3A–3F) in triplicate calf infections. One of the BGF-T20HF oocyst samples used resulted in delayed onset of diarrhea which is responsible for the large SD bars for the average daily fecal volume graph (Fig 3A);

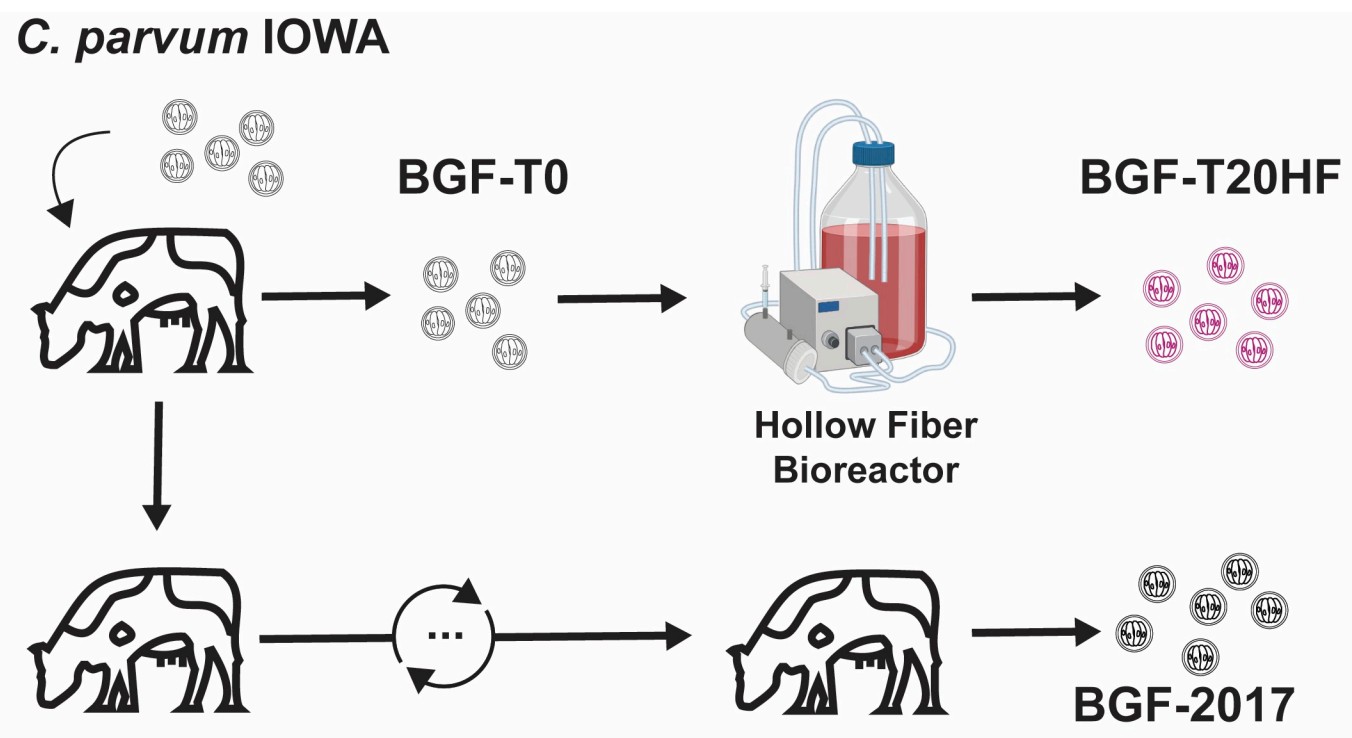

**Fig 2. Strain passage and designation diagram.** *Cryptosporidium parvum* IOWA oocysts, designated, BGF-T0 were purchased from Bunch Grass Farms in March 2016 and sequenced by BGI. *C. parvum* IOWA oocysts designated BGF- 2017 were purchased from Bunch Grass Farms and sequenced by the GGBC at the University of Georgia. BGF-T20HF–Originated from BGF-T0 following 20 months of continuous culture in the HFB and sequenced at BGI. IOWA-ATCC–Genomic DNA was ordered from ATCC (catalog number ATCCPRA-67DQ) and sequenced at the WSI as in (*20*). IOWA-2017 was produced at the *Cryptosporidium* Production Lab (University of Arizona).

however, the sum of the total fecal volumes for each group showed no significant differences (Fig 3B). The daily oocyst shedding by calves infected with BGF-T20HF, BGF-2017 and IOWA-2017 were determined daily from 3–10 days PI (Fig 3C). The statistical difference in the number of oocysts shed in the different infected groups was subjected to Kruskal Wallis analysis which indicated the p-value (0.13) was greater than the significance level ($\alpha$ = 0.05), hence the difference between the mean ranks of all groups was not big enough to be statistically different. This conclusion was supported by the test statistic (H = 4.0808) which is in the 95% region of acceptance [0, 5.9915] supporting the conclusion that there is no significant difference between the mean ranks of any pair (Fig 3C). This conclusion was supported using Mann Whitney analysis of the average total oocysts shed for BGF-T20HF compared to BGF-2017 using a 1 tail test with a *p* of 0.01 resulted in a U value of 18 (critical value of U at *p* <0.01 is 9), and a z-score of -1.41778 (*p* value is 0.0778) indicating there was no significant difference between the data obtained for BGF-T20HF and BGF-2017. Mann Whitney analysis of BGF-T20HF compared to IOWA-2017 confirmed the lack of statistical difference between the hollow fiber cultured oocysts and animal passaged oocysts, resulting in a U value of 19 and a z-score of -1.31276. Fecal consistency was scored from 1–4 (Fig 3D) with higher scores representative of more fluid feces (four being the highest score and one the lowest). One calf infected with the BGF-T20HF oocysts had delayed onset diarrhea causing the dip at day 4 PI which resulted in the differences observed between BGF-T20HF and IOWA-2017 at days 6 and 7 PI.

Daily clinical evaluation scores (lower score values indicate healthier calves) of infected versus control calves was performed and the means for each group of 3 calves determined which revealed that BGF-T20HF and the parent isolate BGF-2017 did not statistically differ (Fig 3E).

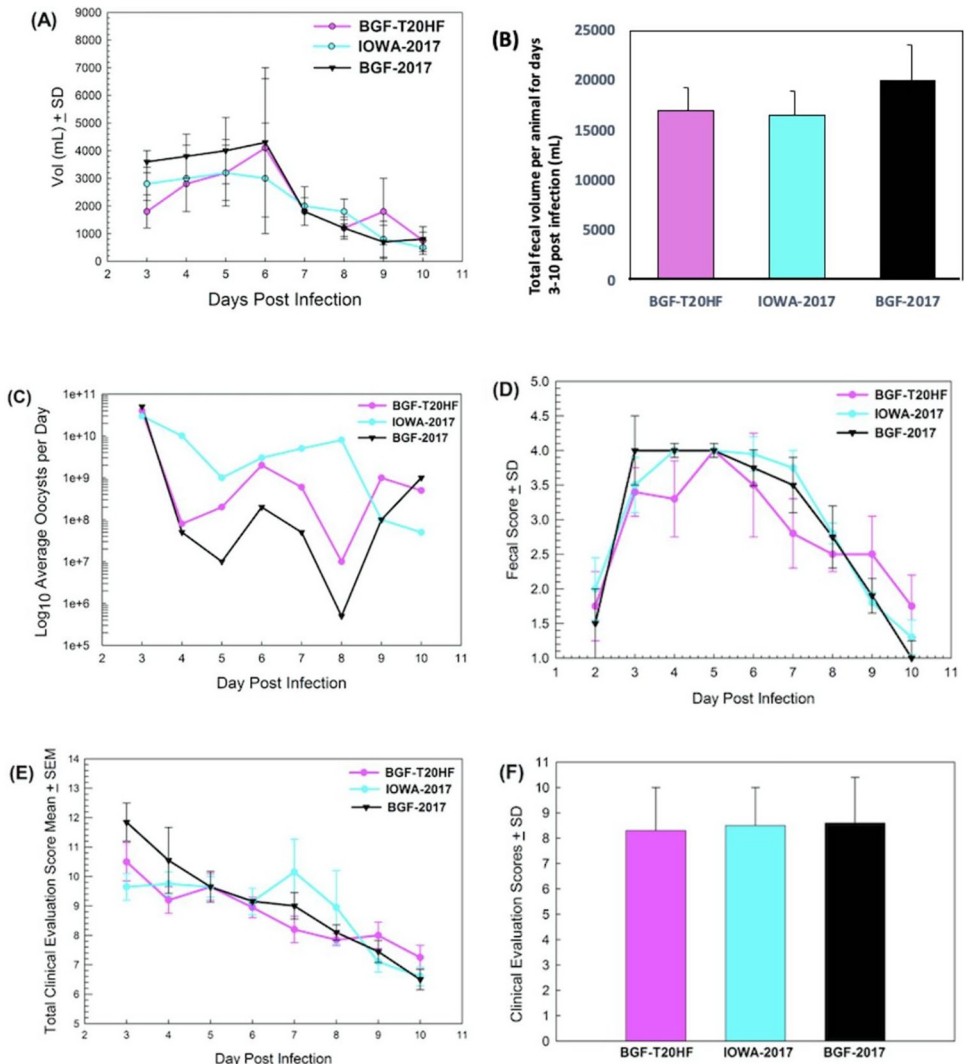

**Fig 3. Infectivity and clinical scores for *C. parvum* isolates BGF-T20HF, compared with calf-passaged isolates BGF-2017 and IOWA-2017.** (A) Average daily fecal volume per calf ± SD of the number of calves in parenthesis BGF-T20HF (4), BGF-2017 (4), IOWA-2017 (5). One HFB (hollow fiber bioreactor derived oocysts) calf was delayed in onset of diarrhea, hence reason for high SDs. (B) Average total fecal volume per calf ± SD of the number of calves in parenthesis BGF-T20HF (4), BGF-2017 (4), IOWA-2017 (5). (C) Daily oocyst numbers shed by BGF-T20HF, BGF-2017, and IOWA-2017. The hollow fiber cultured isolate BGF-T20HF had a similar shedding profile to the parent isolate, BGF-2017 from days 3–10. Non-parametric statistical analysis by the Kruskal-Wallis H-test revealed that there were no significant differences between the groups. (D) Fecal consistency BGF-2017, BGF-T20HF, and IOWA-2017. (E) Daily clinical evaluation. Lower score values indicate healthier calves. Onset of diarrhea was delayed in one BGF-T20HF infected calf. (F) Clinical evaluation score means.

Further, the average of the total clinical evaluation scores for days 3–10 PI indicate no statistical differences between all isolates used in the study (Fig 3F). Cumulatively, the results obtained from the *in vivo* data indicate there is no changes in the infection sequelae of the HFB cultured parasites, BGF-T20HF, in the calf model after 20 months of continuous *in vitro* growth.

## Genome sequence data

Genome sequences were generated at BGI from ~1 µg of DNA using the 20-month HFB *in vitro* cultured parasites (BGF-T20HF) and 1µg of DNA from the parental BGF-T0 used to

**Table 1. Classification of the variant effects observed\* in all BGF samples when compared to *C. parvum* IOWA-ATCC.**

| Variant effect | BGF-T0 | BGF-T20HF | BGF-2017 |
|---|---|---|---|
| 3′ UTR variant | 1 | 0 | 0 |
| 5′ UTR variant | 1 | 1 | 2 |
| Conservative in-frame deletion | 2 | 1 | 2 |
| Conservative in-frame insertion | 5 | 1 | 2 |
| Disruptive in-frame deletion | 6 | 3 | 11 |
| Disruptive in-frame insertion | 1 | 2 | 0 |
| Downstream gene variant | 15 | 7 | 10 |
| Frameshift variant | 10 | 2 | 10 |
| Frameshift variant & stop gained | 1 | 1 | 1 |
| Non-synonymous variant | 20 | 12 | 5 |
| Splice donor variant & intron variant | 1 | 1 | 1 |
| Splice region variant & intron variant | 2 | 0 | 2 |
| Synonymous variant | 7 | 10 | 1 |
| Upstream gene variant | 50 | 43 | 64 |
| Total number of variants | 122 | 84 | 111 |

\*Tables B-D in S1 Table

initiate the HFB culture. The generated sequences were then compared to a new genome sequence assembly designated *C. parvum* IOWA-ATCC [23] in conjunction with a genome sequence for BGF-2017 (supplied by Boris Striepen). Genome-wide, comparison of variants revealed that BGF-T20HF contains less variation overall, except for synonymous substitutions (Table 1 and Fig 4A). Analysis of deletion (D) and insertion (I) events revealed (44 D, 38 I) for the parental BGF-T0, (25 D, 21 I) for BGF-T20HF, and (62 D, 29 I) for BGF-2017 relative to the *C. parvum* IOWA-ATCC genome sequence (Table B in S1 Table). BGF-T20HF did not show copy number variation of genes or genome segments when compared to BGF-T0 (S1 Fig). Comparative analyses of the coding regions (CDSs) from these 4 genome sequences revealed that BGF-T0, BGF-T20HF and BGF-2017 contain 7, 10 and 1 synonymous (S) and 20, 12 and 5 non-synonymous (N) substitutions respectively, relative to *C. parvum* IOWA-ATCC (Table 1). Surprisingly, more non-synonymous substitutions are observed in all strains. Overall, sequence analysis indicates a less diverse population and only minor differences in variation between BGF-T20HF relative to parasites passaged in calves (BGF-T0, BGF-2017, IOWA-ATCC) (Tables 1 and B-E in S1 Table).

When examining CDSs, low levels of variation are detected in all strains and some variants are shared. Variants that change the CDS are enumerated in Fig 4B and Table F in S1 Table. Most of the BGF- T20HF CDS variants were found to be outside of predicted active site or major domain areas (Table C in S1 Table) and are most often associated with non-cytoplasmic regions and coils. However, two variants unique to BGF-T20HF were found to have a missense or disruptive variant that resulted in CDS changes in a predicted InterProScan protein domain. The gene CPATCC_0021250 which encodes a protein phosphatase inhibitor has a mutation (p.Thr32 Asn34del) and CPATCC_0012170 which encodes a P-type ATPase like protein has a missense variant (p.Val12Leu) in a predicted cation-transporting domain. Analysis of the parental BGF-T0 also revealed three variants of putative lesser impact that are not detected in BGF-T20HF or the other strains. These variants are located in CPATCC_0037840, CPATCC_0032770 and CPATCC_0023800 (see Tables B and E in S1 Table).

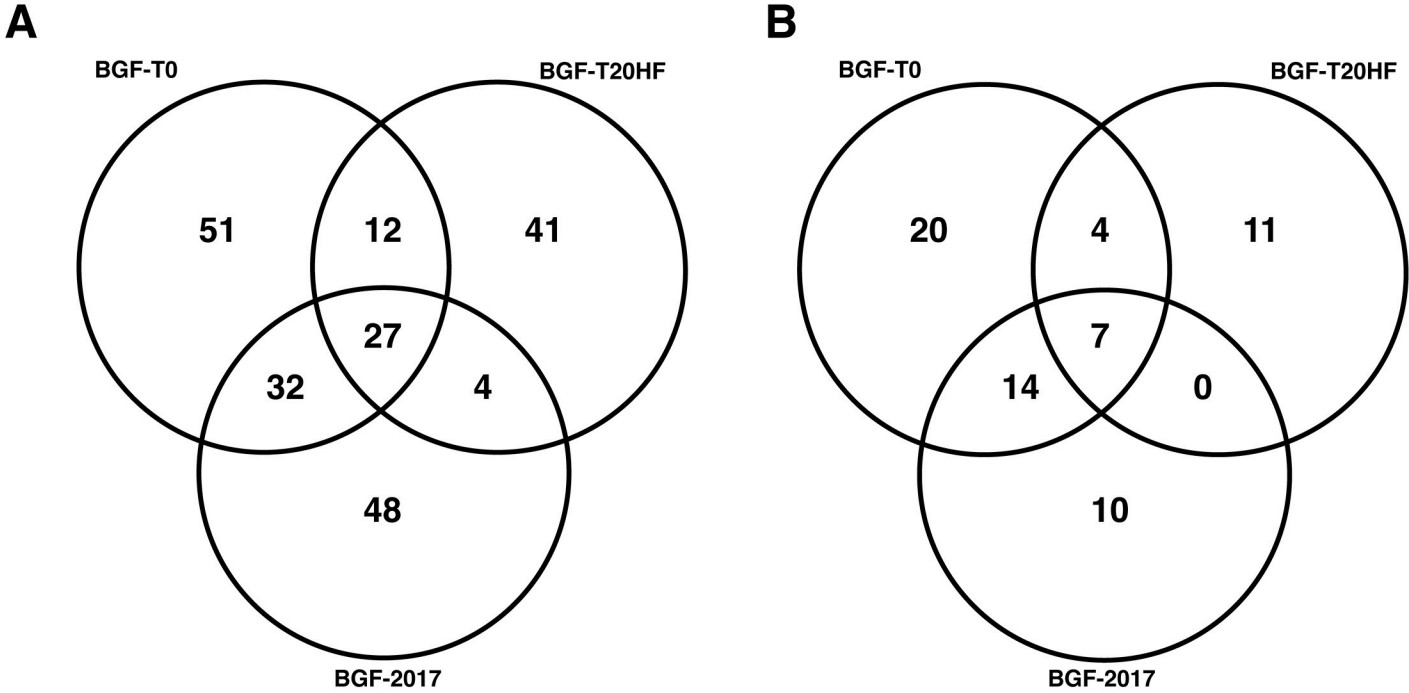

**Fig 4. Venn diagram of observed Genome-wide variation found. (A)** All called variants; and **(B)** Moderate to high-impact variants found in protein coding regions. Data are located in Tables E and F in S1 Table.

### Variants of lesser significance detected in BGF-T0 only

Analysis of the parental BGF-T0 also revealed three variants of putative lesser impact that are not detected in BGF-T20HF or the other strains. These variants are in CPATCC_0037840 which encodes an extracellular membrane protein with a signal peptide which had Ser[336], a polar uncharged amino acid changed to the hydrophobic residue Leu[336] in the third loop (amino acids 323–343) of the 9-loop transmembrane domain. It is unlikely however, that this change will have a significant effect on the function of the transmembrane loop. CPATCC_0032770 which encodes casein kinase had a single amino acid change at position Thr[219] where the polar uncharged amino acid threonine was replaced by a hydrophobic residue, Isoleucine in the protein kinase domain (amino acids 9–278). Finally the polar, uncharged residue, Ser[4] was replaced by the hydrophobic residue Phe[4] in the signal peptide domain MRNSVILKIILFSFLDLIYS of the cysteine-rich secretory protein, CPATCC_0023800, which does not affect the signal peptide and is outside of the SCP domain. Overall it is our conclusion that the observed changes are not in critical domains and are unlikely to affect peptide function.

### Discussion

The use of hollow fiber biotechnology has been successfully employed for the *in vitro* growth of all parasite life cycle stages of *Plasmodium falciparum* [42,43], and *C. parvum* [8]; it has also been shown to have great potential for the mass production of specific parasite stages, such as *Plasmodium falciparum* sporozoites for use in vaccine production [44]. The technique enables the development of a 3D culture environment allowing the formation of a polarized apical cell

surface and a basal cell surface for transport of nutrients and oxygen to the host cells, reproducing the cellular environment found in host tissues. There have been many recent advances in the *in vitro* cultivation of *C. parvum* which permit access to laboratory cultured parasite stages such as the organoid culture method [45,46], that permits access to parasite stages for morphological and molecular analysis. The advantage of the hollow fiber bioreactor is that it provides a method to produce large quantities of *in vitro* cultured *C. parvum* oocysts that are free from the harsh chlorination treatment necessary for the sterilization of animal generated oocysts. Parasites generated using this method have several advantages over animal generated parasites notably they provide a novel method for the generation of parasite stages needed for vaccine development, which has shown to have great promise for a malaria vaccine [44]. *In vitro* cultured parasites permit extended (beyond 48 h) drug testing protocols to be employed and provide a method for preliminary pharmacokinetic/pharmacodynamic data to be obtained for candidate chemotherapeutic agents [47,48]. Currently no other *in vitro* culture method provides access to sufficient oocyst numbers ($10^7$ to $10^8$ oocysts/mL) that can be generated in a GLP facility for use in human clinical trials. However, long term *in vitro* culture of parasites can result in loss of virulence factors and subtle genetic variation that impact their ability to be useful models for chemotherapeutic trials [17–20]. For these reasons we evaluated *in vitro* cultured *C. parvum* after 20 months of continuous culture in the HFB for virulence and genomic changes. We found minor differences in the virulence of *C. parvum* BGF-T20HF compared to either the parent IOWA isolate BGF-2017 or the University of Arizona isolate IOWA-ATCC.

*C. parvum* IOWA oocysts, as a population and not a cloned strain, have been propagated in bovine and murine models in multiple locations since the late 1970's. Molecular divergence was observed when examining 4 loci from 19 samples collected in 2006 from different locations or at different times with some samples being more similar than others [49]. Thus, the number of whole genome differences detected here among the 4 isolates examined (regardless of mode of propagation) is not surprising. However, it serves as a reminder to the community of the need for careful sample naming, lineage tracking, and sequence naming given changes that occur with time via drift and the generation of variants during replication and mutational events.

In conclusion, *in vitro* cultures of *C. parvum* using the hollow fiber bioreactor for a period of 20 months produced parasites that had no decrease in virulence properties, had similar clinical scores in a calf model, and demonstrated similar patterns of genomic variation as found within animal-cultured parasites. Hence the HFB permits extended *in vitro* (beyond 48 h) drug testing protocols to be employed, in addition to preliminary pharmacokinetic/pharmacodynamic data to be obtained which is currently only available using animal models. In addition, because *in vitro* cultured parasites do not require treatment with chlorine-based reagents to remove bacteria and other gut flora they provide a more controlled and GLP amenable source of oocysts for use in clinical trials.

## Supporting information

**S1 Fig. Read depth analysis of BGF-T0 and BGF-T20HF.** Sequence reads were aligned to *C. parvum* IOWA-ATCC genome assembly to assess deletions, insertions, and duplications. Both sample read depths were normalized by the whole genome average depth to get the estimated copy number across all chromosomes.
(JPG)

**S1 Table.** Table A. Sample Summary. Table B. BGF-T0 variants. Table C. BGF-T20HF variants. Table D. BGF-2017 variants. Table E. Shared and unique variants. Table F. Variants of

putative effect.
(XLSX)

## Acknowledgments

The authors acknowledge Dr. Stephen Ward (Bill and Melinda Gates Foundation) for helpful discussions in the planning and execution of this research. Beijing Genomics Institute at Shenzhen, China for Next-Gen sequencing of the *C. parvum* genome (BGF-T0 and BGF-T20HF).

## Author Contributions

**Conceptualization:** Nigel Yarlett, Michael W. Riggs, Jessica C. Kissinger.

**Data curation:** Mary Morada, Deborah A. Schaefer, Elizabeth Carranza, Rodrigo de Paula Baptista, Michael W. Riggs, Jessica C. Kissinger.

**Formal analysis:** Nigel Yarlett, Deborah A. Schaefer, Elizabeth Carranza, Rodrigo de Paula Baptista, Jessica C. Kissinger.

**Funding acquisition:** Nigel Yarlett, Michael W. Riggs, Jessica C. Kissinger.

**Investigation:** Nigel Yarlett, Mary Morada, Deborah A. Schaefer, Kevin Ackman, Elizabeth Carranza, Rodrigo de Paula Baptista.

**Methodology:** Nigel Yarlett, Mary Morada, Deborah A. Schaefer, Rodrigo de Paula Baptista.

**Project administration:** Nigel Yarlett.

**Resources:** Jessica C. Kissinger.

**Supervision:** Nigel Yarlett, Deborah A. Schaefer, Michael W. Riggs, Jessica C. Kissinger.

**Validation:** Nigel Yarlett, Mary Morada, Michael W. Riggs, Jessica C. Kissinger.

**Visualization:** Jessica C. Kissinger.

**Writing – original draft:** Nigel Yarlett.

**Writing – review & editing:** Nigel Yarlett, Deborah A. Schaefer, Rodrigo de Paula Baptista, Michael W. Riggs, Jessica C. Kissinger.

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
