## [Decision Letter · Decision Letter 0]

2 Aug 2023

Dear Dr Yarlett

Thank you very much for submitting your manuscript "Genomic and virulence analysis of in vitro cultured Cryptosporidium parvum" for consideration at PLOS Pathogens. As with all papers reviewed by the journal, your manuscript was reviewed by members of the editorial board and by several independent reviewers. The reviewers appreciated the attention to an important topic. Based on the reviews, we are likely to accept this manuscript for publication, providing that you modify the manuscript according to the review recommendations.

As you can see, I have received diametrically different opinions from the reviewers. The opinion of two of the reviewers is to accept, while the third is to reject. The main criteria for rejection is that the use of the hollow fiber bioreactor for culture of Cryptosporidium parvum has already been published in various papers and, therefore, is not "new" . However, I believe that the demonstration of the genomic stability of the parasite during culture and the preservation of its capacity to infect mice and calve during culture are important. Therefore, I am recommending its publication with minor revisions. For this paper, I would not require the identification and validation of specific factors in virulence.

Please modify the paper to address the specific concerns raised by the reviewers and mine.

Editor minor revisions:

Please give more details of the sequencing protocol use (for example only short reads or also long reads) and the depth of coverage obtain for each condition

Please indicate if there is a statistical difference in the amount of oocysts shed in the different infected groups when the data for the three groups is tested (perhaps a Kruskal Wallis test) or if the statistical difference only is evident when individual groups are compared with each other. It appears that the pattern of shedding between BGFT20 and BGF-2017 is very similar and that the peak shedding of Iowa is latter.

Please correct graph 3C (or text) as the number of days in the text differs from the number of data points shown in the graph.

Sincerely,

Ana Maria Cevallos, Ph.D.

Guest Editor

PLOS Pathogens

P'ng Loke

Section Editor

PLOS Pathogens

Kasturi Haldar

Editor-in-Chief

PLOS Pathogens

orcid.org/0000-0001-5065-158X

Michael Malim

Editor-in-Chief

PLOS Pathogens

orcid.org/0000-0002-7699-2064

As you can see, I have received diametrically different opinions from the reviewers. The opinion of two of the reviewers is to accept, while the third is to reject. The main criteria for rejection is that the use of the hollow fiber bioreactor for culture of Cryptosporidium parvum has already been published in various papers and, therefore, is not "new" . However, I believe that the demonstration of the genomic stability of the parasite during culture and the preservation of its capacity to infect mice and calve during culture are important. Therefore, I am recommending its publication with minor revisions. For this paper, I would not require the identification and validation of specific factors in virulence.

Please modify the paper to address the specific concerns raised by the reviewers and mine.

Editor minor revisions:

Please give more details of the sequencing protocol use (for example only short reads or also long reads) and the depth of coverage obtain for each condition

Please indicate if there is a statistical difference in the amount of oocysts shed in the different infected groups when the data for the three groups is tested (perhaps a Kruskal Wallis test) or if the statistical difference only is evident when individual groups are compared with each other. It appears that the pattern of shedding between BGFT20 and BGF-2017 is very similar and that the peak shedding of Iowa is latter.

Please correct graph 3C (or text) as the number of days in the text differs from the number of data points shown in the graph.

Reviewer Comments (if any, and for reference):

Reviewer's Responses to Questions

**Part I - Summary**

Reviewer #1: This is a straightforward study to examine if any significant changes occur in C. parvum cultured in vitro in a hollow fiber system develop during 20 months of continuous culture. This strain (BGF-T20HF) is compared to the original seeding strain (BGF-10) and to the same strain that was maintained in cows (in vivo), i.e., BGCF-2017. While minor changes were noticed in the BGF-T20HF genome sequence data, there were similar types of changes in BGF-2017 and in BGF-10 suggesting that in vitro culture over this period of time did not significantly alter Cp. In addition, clinical infection of calves (although only 3 per strain) did not show any major changes in virulence or ability of the prolonged in vitro isolate to cause clinically relevant disease; as compared to the starting strain or the strain maintained in cows. Overall, this data provides important validation of the culture system and the use of parasites from this system for scientific investigations.

Reviewer #2: In vitro culture of Cryptosporidium has historically been one of the challenges that has limited development of effective therapeutics for this important intestinal parasite that afflicts a large number of people globally, particularly children in low-resource settings. The report in 2016 by the Yarlett lab of successful culture of Cryptosporidium in hollow fiber bioreactors (HFB) offered some hope that a novel technology enabling the culture of large numbers of parasites and accurate testing of the pharmacokinetics of chemotherapeutic candidates could become available. The current manuscript extends the findings of the 2016 paper and demonstrates comparable infectivity, virulence, and genomic stability of parasites cultured in HFB (in comparison to culture using standard passage in calves) using one of the main clinically relevant models for cryptosporidiosis, infection of newborn calves.

Overall, this is a well-executed study that generated quality data and makes conclusions that are consistent with the data presented. Assuming the minor revisions suggested are made, I would support publication of the manuscript.

Reviewer #3: The authors have utilized the hollow fiber bioreactor technology to investigate the virulence properties of Cryptosporidium parvum after 20 months within this device.

There are no major flaws in the experimental design, but the novelty is quite weak. The manuscript doesn't convince me about the outputs of the whole study and doesn't provide any novelty other than advertising the use of the HFB system.

**Part II – Major Issues: Key Experiments Required for Acceptance**

Reviewer #1: None

Reviewer #2: None required.

Reviewer #3: All of the results (potential proteins that are affected) are based on predictions. I would have like to see either localisation or biochemical characterisation of these proteins to convince the readers about the impact of these proteins for virulence.

**Part III – Minor Issues: Editorial and Data Presentation Modifications**

Reviewer #1: The others cite unpublished data on T. gondii in these culture system. If a manuscript has been submitted to a preprint service such as bioRxiv then this should be provided in the references.

Reviewer #2: Page 5, line 17 (Introduction)

The statement that NTZ is inadequate in children under 3 years old is inaccurate. Reference #5 (Amadi et al. 2002 Lancet) found NTZ was modestly effective in HIV-negative children, but not HIV-positive children. I suggest instead highlighting here that NTZ is not approved for use in children under 12 months of age, which is the most population most vulnerable to the effects of cryptosporidiosis.

Page 13, line 14 (Calf infections)

The sentence "The daily oocyst numbers shed indicate that BGF-T20HF is statistically significant compared to BGF-2017 and IOWA-2017 for days 3 and 10 PI" is confusing and needs to be reworded. It is unclear if the authors mean to say oocyst shedding from calves infected with BGF-T20HF is statistically significantly greater or less than BGF-2017 and IOWA-2017. Furthermore, in Figure 3C, it appears that BGF-T20HF is virtually identical to BGF-2017 on day 3 PI, so it is hard to understand how these values could be statistically significantly different. Finally, there is no data shown for day 10 PI, despite this data being referenced in the text and the figure legend.

Page 13, line 15 (Calf infections)

The sentence "All other days are significantly higher for BGF-T20HF (days 5-9 PI) and IOWA-2017 (days 4-9 PI) compared to BGF-2017 (Fig 3C)" is not consistent with the data shown in Fig 3C. For example, on day 8 PI, BGF-2017 appears to be virtually identical to IOWA-2017. Furthermore, on day 9 PI, BGF-2017 is greater than BGF-T20HF, contrary to what is stated.

Page 14, line 1 (Calf infections)

Interpretation of the fecal consistency data (and reference to Fig 3D where it is shown) are missing. I suggest moving text from figure legend to the main body here (see comment in Fig 3D legend).

Page 14, line 3 (Calf infections)

The explanation "Onset of diarrhea was delayed in one BGF-T20HF infected calf (Fig 3F)" appears to be superfluous because the data in Figs 3E-F appear to be very similar for BGF-T20HF as compared to BGF-2017 and IOWA-2017.

Page 16, line 22 (Discussion)

The reference after the sentence, "Parasites generated using this method...have great promise for a malaria vaccine" should be reference #43 (Eappen et al. 2022 Nature), not reference #44.

Page 27, line 12 (Figure 1 legend)

Should be "HCT-8 cells", not simply "HCT cells".

Page 28, line 10 (Figure 3 legend)

I suggest that the two interpretive sentences ("One calf infected...due to this delay.") are moved from the figure legend to the main body of the text where Fig 3D is described, as suggested above.

Page 30 (Figure 2)

IOWA-2017 is referenced in the figure legend and elsewhere in the manuscript, but not shown in the figure. I recommend clarifying where IOWA-2017 fits in this schematic.

Page 31 (Figure 3)

The y-axis of Fig 3B is confusing because it uses scientific notation (e.g., "1e+4"), yet the scale is linear, not logarithmic (unlike Fig 3C, which is on a log scale). I suggest using standard notation for the y-axis (10 L, 20 L, 30 L, etc.) for Fig 3B.

Reviewer #3: - Figure 1 is the typical figure that has been presented in all of the papers of the same group when they published HFB-related work.

- Supplementary figure 1. I do not understand the purpose of this experiment at all. And even if they manage to convince me about the purpose, their interpretation is also wrong since the bands (VspI samples) are not of the same size. Also, the quality of the figure is poor.

- Cryptosporidia is not a latin name and it should not be in italics.

PLOS authors have the option to publish the peer review history of their article (what does this mean?). If published, this will include your full peer review and any attached files.

Reviewer #1: No

Reviewer #2: No

Reviewer #3: No

Figure Files:

Data Requirements:

Reproducibility:

References:

---

## [Decision Letter · Decision Letter 1]

14 Nov 2023

Dear Dr Yarlett

Thank you very much for submitting your manuscript "Genomic and virulence analysis of in vitro cultured Cryptosporidium parvum" for consideration at PLOS Pathogens.

The concerns of the majority of reviewers have been addressed.  However, there are still corrections as described by reviewer two that need to be addressed.

Sincerely,

Ana Maria Cevallos, Ph.D.

Guest Editor

PLOS Pathogens

P'ng Loke

Section Editor

PLOS Pathogens

Kasturi Haldar

Editor-in-Chief

PLOS Pathogens

orcid.org/0000-0001-5065-158X

Michael Malim

Editor-in-Chief

PLOS Pathogens

orcid.org/0000-0002-7699-2064

THere are minor corrections as described by reviewer two that need to be addressed.

Reviewer Comments (if any, and for reference):

Reviewer's Responses to Questions

**Part I - Summary**

Reviewer #1: This is a straightforward manuscript that describes the effect of long term in vitro culture of C. parvum in the hollow fiber system on the pathogenicity of this organism as well as the molecular biology (gene mutations) from this prolonged (20 month) cultivation. This type of work is important as it establishes that parasites from prolonged cultivation are the same as those obtained from animal passage. This is important information to support the use of the in vitro cultured parasites for studies such as human (or animal ) challenge, vaccine development and genetic modification.

Reviewer #2: Most of the revisions from the previous version are satisfactory. However, there are a few corrections that still need to be made, as described in the attached PDF and Part III below.

Reviewer #3: The authors have utilized the hollow fiber bioreactor technology to investigate the virulence properties of Cryptosporidium parvum after 20 months within this device.

There are no major flaws in the experimental design, but the novelty is quite weak. The manuscript doesn't convince me about the outputs of the whole study and doesn't provide any novelty other than advertising the use of the HFB system.

**Part II – Major Issues: Key Experiments Required for Acceptance**

Reviewer #1: None the authors have addressed previous reviewer concerns.

Reviewer #2: None.

Reviewer #3: - Figure 1 is the typical figure that has been presented in all of the papers of the same group when they published HFB-related work.

**Part III – Minor Issues: Editorial and Data Presentation Modifications**

Reviewer #1: In the discussion, page 15 line one the correct reference for in vitro production of P. falciparum sporozoites is [42] not [43]

Reviewer #2: Please see the comments in the attached PDF. Note I have made comments in the "Revised article with changes highlighted" version of the manuscript at the end of the file.

Specifically, the two most critical issues that need to be addressed are:

1. Revision of Figs 3A and 3B to ensure total fecal volumes are consistent.

2. Revision of the Fig 3C legend to align with the changes in the main text describing the Fig 3C results.

Reviewer #3: No further comments

---

## [Editor Report · Decision Letter 2]

22 Jan 2024

Dear Dr. Yarlett,

We are pleased to inform you that your manuscript 'Genomic and virulence analysis of in vitro cultured Cryptosporidium parvum' has been provisionally accepted for publication in PLOS Pathogens.

Best regards,

Ana Maria Cevallos, Ph.D.

Guest Editor

PLOS Pathogens

P'ng Loke

Section Editor

PLOS Pathogens

Kasturi Haldar

Editor-in-Chief

PLOS Pathogens

orcid.org/0000-0001-5065-158X

Michael Malim

Editor-in-Chief

PLOS Pathogens

orcid.org/0000-0002-7699-2064

The remaining corrections have been made. Thank you
---

## [Editor Report · Acceptance letter]

22 Feb 2024

Dear Dr. Yarlett,

We are delighted to inform you that your manuscript, "Genomic and virulence analysis of in vitro cultured Cryptosporidium parvum," has been formally accepted for publication in PLOS Pathogens.

Best regards,

Michael Malim

Editor-in-Chief

PLOS Pathogens

orcid.org/0000-0002-7699-2064